# Paroxysmal sympathetic hyperactivity during neurorehabilitation for severe acquired brain injury: current Scandinavian practice and Delphi consensus recommendations

Alison K Godbolt [1], Alexandros Zampakas [2], Catharina Nygren Deboussard [1]

¹Department of Clinical Sciences, Danderyd Hospital, Karolinska Institute, Stockholm, Sweden
²Department of Rehabilitation Medicine, Danderyd University Hospital, Stockholm, Sweden

**Correspondence to**
Alison K Godbolt;
alison.godbolt@ki.se

## ABSTRACT

**Objectives**  To document current practice and develop consensus recommendations for the assessment and treatment of paroxysmal sympathetic hyperactivity (PSH) during rehabilitation after severe acquired brain injury.

**Design**  Delphi consensus process with three rounds, based on the Guidance on Conducting and REporting DElphi Studies (CREDES) guidelines, led by three convenors (the authors) with an expert panel. Round 1 was exploratory, with consensus defined before round 2 as agreement of at least 75% of the panel.

**Setting**  A working group within the Nordic Network for Neurorehabilitation.

**Panel participants**  Twenty specialist physicians, from Sweden (9 participants), Norway (7) and Denmark (4), all working clinically with patients with severe acquired brain injury and with current involvement in clinical decisions regarding PSH.

**Results**  Consensus was reached for 21 statements on terminology, assessment and principles for pharmacological and non-pharmacological treatment, including some guidance on specific drugs. From these, an algorithm to support clinical decisions at all stages of inpatient rehabilitation was created.

**Conclusions**  Considerable consensus exists in the Nordic countries regarding principles for PSH assessment and treatment. An interdisciplinary approach is needed. Improved documentation and collation of data on treatment given during routine clinical practice are needed as a basis for improving care until sufficiently robust research exists to guide treatment choices.

## INTRODUCTION

Following severe acquired brain injury, some patients develop episodes of excessive sympathetic activity that can be very pronounced, with tachycardia, tachypnoea, fever, motor posturing, sweating and hypertension. The 2014 publication[1] of consensus on definitions, nomenclature and diagnostic criteria was a major step forward regarding this syndrome that had previously been referred to by a very

---

### STRENGTHS AND LIMITATIONS OF THIS STUDY

⇒ Robust process for forming consensus according to published recommendations.
⇒ Involvement of clinicians working at all stages of inpatient rehabilitation.
⇒ Panel representation from three Scandinavian countries and most healthcare regions within countries.
⇒ Insufficient number of panel participants with experience of care after discharge from inpatient rehabilitation for consideration of this phase.
⇒ Panel participants required to have clinical experience of paroxysmal sympathetic hyperactivity during the past year, but no specific requirement on number of patients treated by panel participants.

---

large number of different terms. Paroxysmal sympathetic hyperactivity (PSH) as defined in 2014 has become the recommended and established term for this syndrome. Similar symptoms may also occur due to a broad range of other complications of brain injury or comorbidities, including but not limited to infections, heterotopic ossification, alcohol or drug withdrawal. Alternative or contributing diagnoses must be identified and (when necessary) treated. A diagnosis of PSH is made with consideration of the clinical features above, together with diagnostic likelihood after consideration of differential diagnoses.[1]

Subsequent studies[2] have shown an association of PSH with longer length of hospital stay and with more unfavourable outcomes. Experts in neurocritical care have published articles[2,3] suggesting treatment options, but a recent scoping review[4] found a lack of evidence to inform clinical decision making when treating patients with PSH after severe acquired brain injury.

Mechanisms underlying PSH remain incompletely understood and are likely to be complex. One prominent model is the excitatory/inhibitory ratio model,[2 5] whereby impairment of central descending inhibition of sympathetic and motor pathways leads to loss of control of the excitatory action of spinal circuits. This explains pathological sympathetic responses to minimally or non-nociceptive stimuli, such as tracheal suction or placement of orthotics. A central autonomic network subserves this central descending inhibition, comprising both cortical and subcortical components.[6] While certain structures (midbrain, pons, periventricular white matter and corpus callosum[2]) seem to have a key role in PSH, it is the overall severity of brain injury (rather than specific anatomical areas of injury) that likely gives an increased risk of PSH. Increased catecholamine (epinephrine, norepinephrine and dopamine) and ACTH levels have been demonstrated in traumatic brain injury patients during PSH episodes,[7] adding support to theories of sympathetic over-activity. Other mechanisms have also been proposed, although with only limited underlying research: rat studies have suggested putative roles of neutrophil extracellular traps (traumatic brain injury model[8]), and hydrogen sulfide (cerebral infarct model[9]). Published treatment suggestions[2 4] are based to varying extents on targeting of such putative mechanisms.

Many studies and reviews of PSH have originated from authors working within neurointensive care.[2 3] While acknowledging the value of these publications, there is also a need for rehabilitation considerations and the rehabilitation care process to be considered more explicitly to support optimal care continuing into care settings following neurointensive care, for those patients with continuing PSH later after injury. Sedating drugs that may be appropriate for use for patients who are anaesthetised on the neurointensive care unit may become inappropriate as rehabilitation priorities gradually evolve.

In clinical practice, patients may also be transferred between some (or even many) different wards and rehabilitation units during their inpatient care. It is a challenge to create management strategies that are sufficiently flexible to be appropriate on different units with very different staffing and availability of monitoring, and at the same time to be sufficiently clear to support good care when the total number of patients with PSH that individual clinicians in some care settings treat is small. A recent study that is from a neurorehabilitation setting,[10] on patients with Prolonged Disorders of Consciousness (the most severely injured subgroup of all with PSH), noted a prevalence of PSH of 31% in this subgroup at rehabilitation admission, suggesting that PSH is a not inconsiderable issue.

Consensus recommendations have a place in supporting best clinical practice for today's patients, until adequate research evidence can be produced. A Delphi process is an established, structured technique for the formation of consensus based on four methodological aspects:[11] a group of experts ('the panel') answer questions, the process is anonymous to ensure that less dominant views come forward, the process is iterative, with several rounds of questions, and subsequent rounds are based on a summary of group responses to the previous round.

The aim of the project was to develop Scandinavian consensus recommendations for assessment and treatment of PSH during rehabilitation after severe acquired brain injury.

## METHODS

The CREDES recommendations[11] for Delphi processes were used as the basis for the consensus process.

### Justification for use of a Delphi process

Uncertainties regarding best treatment for PSH and the lack of adequate research findings to assist physicians in clinical decisions were discussed at a meeting of the Brain Injury Pharmacotherapy Special Interest Group of the Nordic Network for Neurorehabilitation[12] at a meeting in Oslo in November 2022. A decision was taken to hold a Delphi process to establish the degree of consensus.

### Planning and process

During the spring of 2023, the convenors (the authors) created a project plan and questions for the first round based on previous discussions of areas of difficulty and published, peer-reviewed articles on PSH. These were discussed with an independent rehabilitation physician and Neurosurgeon (see acknowledgements) from a different region to the convenors, following which adjustments were made to the project plan and questions. Before round 1, Consultant physicians in Rehabilitation Medicine at the authors' department, who were not participants in the panel, answered a pilot version and gave feedback on the question wording and content before final adjustments. The Delphi process was run between August and December 2023; round 1 was distributed in August, round 2 in October and round 3 in December. An anonymised summary of responses was sent to the panel after completion of each round and before distribution of the next round.

### Establishing the panel

Factors of importance regarding choice of panel participants were discussed by the convenors before sending invitations. Representation from each healthcare region in Denmark, Sweden and Norway was sought, with an aim of 1–3 panel participants from each region. We sought panel participants who were physicians who had completed specialist training, worked clinically with patients with severe acquired brain injury in the Nordic countries, and had been involved in clinical decisions about drugs treatment of PSH in the previous year; these formed criteria for participation. Potential panel participants were identified via lists of participants in the Nordic Network of Neurorehabilitation meeting in November 2022, by network contacts, and by contact with the

authors of a recent Scandinavian publication on PSH.[13] Invited participants received an email with information about the project including a reference to the key 2014 publication on consensus criteria, extent of panel involvement and confirmed that they met the above criteria for panel participants prior to answering round 1. Answers were in English or national languages.

## Patient group

Patients with severe acquired brain injury defined as any sudden insult to a previously essentially healthy brain in a patient aged at least 2 years, of a severity that requires initial neurointensive care. Diagnoses include traumatic brain injury, anoxic brain injury, subarachnoid haemorrhage, spontaneous intracerebral haemorrhage and cerebral infarction.

## Rehabilitation focus

The focus of this study was on patients undergoing rehabilitation after severe acquired brain injury, including early rehabilitation which may occur during the later stages of neurointensive care, intermediary care, care on an acute ward or inpatient care on a rehabilitation unit. Due to the focus on rehabilitation, and thereby the need for consideration of promotion of recovery of awareness and cognition, this survey did not include patients who are anaesthetised and mechanically ventilated in the very early stages after brain injury when other treatments may be necessary to minimise secondary brain injury and when such rehabilitation considerations are not yet relevant.

## Definition of consensus

Round 1 was an exploratory round with the aim of finding an optimal focus for the rest of the process. The decision was therefore made (and communicated to panel participants) before round 1 that criteria for consensus would be decided after completion of round 1. Panel participants received the following information on stopping criteria before they answered round 2: 'Statements that 75% of the panel agree with will be considered to represent consensus in the Scandinavian countries (Denmark, Sweden, Norway) at this time. The conveners will consider the development of responses and the likelihood of coming to consensus in specific aspects after round 3, and then consider whether further rounds (maximum 5) will give added value. If after three rounds we have not reached consensus for some items, and it is unlikely that consensus will be reached, areas where at least 50% of the panel agree will also be reported as areas of considerable agreement (but not consensus)'.

## Procedure for the rounds

Round 1 included questions on panel participants' specialty and clinical experience, and on a wide range of aspects of PSH assessment and management. Questions were in various forms, where closed questions (possible answers, 'yes, no, I don't know') were complemented with free-text comment boxes. Round 1 also included questions about drugs used for treatment of PSH. A list was given

based on 29 classes of drugs (with examples) reported as being used for PSH in a recent scoping review.[4] For each phase of care that they participated in, panel participants were asked to indicate all drugs they used, also the three drugs most often used and comment on reasons for their use or non-use. Phases of care were included due to expected differences in treatment priorities with care setting and were defined as follows: (1) Breathing spontaneously with no or only light sedation on the neurointensive care unit. The patient may have respiratory support. (2) Breathing spontaneously with no or only light sedation on an intermediate care unit/high dependency unit. The patient may have respiratory support. (3) Cared for on an acute care ward or rehabilitation ward at an acute hospital. The patient may have respiratory support. (4) Cared for on a rehabilitation ward without acute hospital facilities. (5) Cared for outside hospital (home, nursing home).

Round 2: based on collated responses to round 1, a number of statements were created for round 2, which the panel were asked to respond to with 'agree, disagree' or, 'I don't know'. For each statement, they could also comment and suggest changes. The five drugs, which were included in at least 25% of the panels' lists of the three most commonly used drug treatments for PSH at any phase of care, were carried forward to round 2. Panel participants were asked, for each phase of case which they participated in, to indicate how commonly they prescribed these drugs for PSH on a scale of 0 (never) to 50 (for about half of patients) to 100 (always).

Round 3 included reformulated statements for the minority of areas where consensus had not been reached in round 2, with a new possibility to agree or disagree. Round 3 also included possible refinements for three statements where consensus had already been reached in round 2, but where individual panel participants had suggested refinements or improvements in their responses to round 2. Panel participants in round 3 were asked to confirm their agreement or otherwise with the original version of these statements from round 2, state their agreement or otherwise to the adjusted versions of these statements and finally to express a preference for one version.

## Platform

REDCap[14] is a secure web application for building and managing online surveys and databases. REDCap was used to produce and send out the questionnaires for each round, to send prescheduled and extra reminders for submission of answers and also to distribute summaries of findings from each round before the following round. Personalised email reminders were also sent to panel participants who had not responded by the initial deadline, with extensions given of up to a week. Participants received no compensation for their participation.

## Ethics and informed consent

This study, in which participants were physicians contributing with their knowledge and opinions, did not fall with

the scope of studies requiring ethical approval in Sweden (https://etikprovningsmyndigheten.se/en/) and as such ethical approval was not required. After receiving information about the study, panel participants clicked a response to the following question on the digital platform: 'I give my consent to participate in the panel, for storage of my email address and de-identified responses according to GDPR for the duration of the Delphi-process, and for feedback of anonymized responses to the panel during the process. I also consent to publication of study findings'. Possible responses were 'yes' and 'no'. The digital platform was programmed so that the questionnaire was terminated if a 'no' response was given, and access to the consensus questions was only available after a 'yes' response.

## RESULTS

### Panel participants

Of 23 physicians invited to participate, 20 met the criteria, formed the panel and responded to rounds 1 and 2; 19 of these also responded to round 3. Four panel participants worked with patients in Denmark, seven panel participants in Norway and nine in Sweden. Medical specialities of panel participants were Rehabilitation Medicine or equivalent for 14, intensive care medicine for two, Neurosurgery for three and neurology for one. Three of the Rehabilitation Medicine specialists were also specialists in another area: two in neurology, and one in intensive care medicine. Panel participants had a mean of 10.3 years of clinical experience of acquired brain injury (SD 6,7). Panel participants were from two of four Danish healthcare regions, four of five Norwegian healthcare regions and five of six Swedish healthcare regions.

### Panel participants' clinical involvement at the different phases of care

Six panel participants were involved with decisions regarding PSH treatment for patients breathing spontaneously with no or light sedation on the neurointensive care unit, 10 on the intermediate care (or high dependency) unit, 12 during care on an acute or rehabilitation ward at an acute hospital, 9 during care on a rehabilitation ward without acute hospital facilities and 3 for patients cared for outside hospital (at home or in an nursing home). Some panel participants were involved during several phases of care. The number of panel participants with current experience of patients care for outside hospital was considered too few for formation of consensus and this phase of care was not included in subsequent rounds.

### Process

Consensus was reached for many areas in round 2. A third round was run to allow consideration of refinements of some statements and to consider alternatives to two statements where consensus had not been reached in round 2. The project was concluded at the end of round 3.

### Consensus statements

At least 75% of panel participants agreed to the following statements by the end of round 3. Two statements concerned only the later phases of the chain of care. For these statements, only responses from panel participants who worked during the later phases of care were considered when calculating percentage agreement. Following each statement, the number of panel participants who agreed, disagreed or did not know is given and the round after which consensus was reached for the final wording is given.

1. The term PSH is the preferred term for use in clinical practice. The diagnosis should be made according to the consensus definition of Baguley *et al* 2014. As described by Baguley *et al*, PSH may occur after severe acquired brain injury, including but not limited to traumatic brain injury. *Agree 19, Disagree 0, Don't know 1. Consensus after round 2.*

2. Differential diagnoses should be considered and as far as possible excluded and/or treated before making the diagnosis of PSH. *Agree 20, Disagree 0, Don't know 0. Consensus after round 2.*

3. The PSH-assessment measure (PSH-AM) should be used to support the initial clinical diagnosis and management of PSH. In some clinical settings, aspects of the PSH-AM may be difficult to implement, and additional or alternative supporting tools may need to be developed. *Agree 17, Disagree 1, Don't know 2. Consensus after round 2.*

4. PSH will in most cases have developed and been diagnosed during neurointensive care or intermediary care. In those cases where the diagnosis has not been made, the PSH-AM is also recommended for use on acute wards or inpatient rehabilitation units. *Responses only sought from panel participants working on acute wards or inpatient rehabilitation units, n=14: Agree 14, Disagree 0, Don't know 0. Consensus after round 3.*

5. For patients who have a diagnosis of PSH made on the neurointensive or intermediary care ward and where drug treatment for PSH is ongoing when they are discharged, there is a need for follow-up later in the chain of care. The PSH-AM may be used during inpatient rehabilitation as part of follow-up and to guide decisions on tapering of treatment. *Responses only sought from panel participants working on acute wards or inpatient rehabilitation units, n=14: Agree 14, Disagree 0, Don't know 0. Consensus after round 3.*

6. The effectiveness of prophylactic drug treatment for patients with a risk of PSH is unknown. Negative effects of drug treatment are difficult to evaluate in patients with severe acquired brain injury. At the current time, prophylactic pharmacological treatment aimed at preventing PSH in patients who do not have a PSH diagnosis cannot therefore be recommended. *Agree 19, Disagree 0, Don't know 1. Consensus after round 2.*

7. General principles for treatment: choice of PSH drug therapy should take into consideration phase of

recovery, care setting and current medical and rehabilitation priorities such as need to promote recovery of consciousness and cognition. *Agree 20, Disagree 0, Don't know 0. Consensus after round 2.*

8. When initiating treatment of PSH, treatment should aim to treat both sympathetic hyperactivity and to minimise the most prominent symptoms of PSH in the individual patient. *Agree 19, disagree 0, don't know 0. Consensus after round 3.*

9. PSH triggers should be assessed, and measures taken to minimise these before and during any pharmacological treatment. PSH triggers may include urinary retention, constipation, gastric paresis, spasticity, poor positioning (where spasticity may contribute), allodynic responses to poorly fitting orthotics, pressure sores and other causes of pain, environmental factors such as ambient temperature and noise. Units should develop routines for a structured approach to minimisation of triggers. *Agree 20, Disagree 0, Don't know 0. Consensus after round 2.*

10. Supportive treatment (hydration, reduction of fever, correction of electrolytes, nutrition) should be optimised in parallel to pharmacological treatment for PSH. Delays in pharmacological treatment should be avoided in moderate to severe cases. Units should develop routines for a structured approach to optimisation of supportive treatment. *Agree 17, Disagree 1, Don't know 1. Consensus after round 3.*

11. PSH as the treatment indication should be clearly documented in the medical record when prescribing drugs to treat PSH, when transferring patients between units and at discharge. This is particularly important as many drugs used for PSH also have other indications. Treatment response to any current or previous PSH drug treatment should also be documented. Future medication review later in the chain of care should be planned. *Agree 19, Disagree 0. Don't know 1. Consensus after round 2.*

12. Sedating drugs should be minimised wherever possible when treating PSH, due to negative effects on recovery of consciousness and cognition, risk for other negative effects of sedation such as retained respiratory secretions and impact on participation in rehabilitation. This is especially relevant in rehabilitation settings but should also be considered during neurointensive care and intermediary (high dependency) care. For very severe cases, short periods of treatment with sedating drugs may be unavoidable. *After round 3: Agree 18, Disagree 1, Don't know 0. Consensus on a previous version in round 2, with final consensus on this alternative wording in round 3.*

13. The efficacy and relatively favourable side effect profiles of non-selective beta-blockers and Gabapentin make these appropriate for first-line treatment of PSH for most patients after neurointensive care. *After round 3: Agree 17, Disagree 0, Don't know 1, no answer 1. Consensus reached in round 2 and confirmed after rejection of an alternative wording in round 3.*

14. Opiates are often necessary for PSH early after injury due to good effects on many PSH components. However, they should be tapered when possible later during rehabilitation due to negative effects on alertness and cognition. *Agree 20, Disagree 0, Don't know 0. Consensus after round 2.*

15. Use of benzodiazepines primarily for treatment of PSH should be minimised (regarding dose and treatment duration) during phases in recovery when the focus is rehabilitation, due to negative effects on cognition and alertness. Their use may be necessary in the initial treatment of PSH on the neurointensive care unit and/or high dependency unit but they should be tapered when possible. *Agree 20, Disagree 0, Don't know 0. Consensus after round 2.*

16. Botulinum toxin may be needed for treatment of focal spasticity generally after severe acquired brain injury, and specifically for patients where focal or regional spasticity contributes to triggering of PSH episodes. *Agree 19, Disagree 0, Don't know 1. Consensus after round 2.*

17. Enteral baclofen may be necessary for PSH with considerable disturbances of muscle tone, but negative effects on cognition and alertness may contribute to difficulties with participation in rehabilitation. These negative effects should be considered when prescribing and during follow-up. *Agree 20, Disagree 0, Don't know 0. Consensus after round 2.*

18. An intrathecal baclofen pump should be considered for severe PSH where other measures have not been effective. *Agree 16, Disagree 1, Don't know 3. Consensus after round 2.*

19. PSH abates (reduces or disappears) with increased time after injury for most patients. Drug and other treatments for PSH should therefore be reviewed regularly and reduced or ceased if and when possible. *Agree 20, Disagree 0, Don't know 0. Consensus after round 2.*

20. The course of PSH in the postacute phases (on the rehabilitation ward and beyond) and optimal tapering of medication when episodes abate are insufficiently documented and should be a focus of future research and development work. *Agree 15, Disagree 0, Don't know 2, No answer 3. Consensus after round 2.*

21. Ethical challenges in treating PSH in patients who often lack the capacity to consent to treatment are acknowledged. When the patient is unable to participate in treatment discussions next of kin and/or legal representatives should be informed of treatment decisions and involved in discussions of the often difficult balance between negative effects of PSH and potential negative effects of treatment (side effects, sedation). The presence of PSH is also associated with more severe brain injury with poorer outcomes, about which relatives should be informed. *Agree 19, Disagree 1, Don't know 0. Consensus after round 2.*

Table 1 Panel participants' frequency of use of common drugs for PSH

| | Neurointensive care | Intermediary care | Acute hospital—acute or rehabilitation ward | Inpatient rehabilitation outside acute hospital |
|---|---|---|---|---|
| Respondents | n=6 | n=9 | n=9 | n=8 |
| Drug: | | | | |
| Alpha-2 agonists, for example, clonidine | 80.5 (30–100) | 50 (16–100) | 23.5 (0–96) | 10 (0–90) |
| Baclofen (enteral) | 32.5 (0–89) | 26.5 (0–75) | 22 (0–70) | 28 (12–50) |
| Beta-blocker, non-selective, for example, propranolol | 65 (20–94) | 83 (16–100) | 81 (50–100) | 75.5 (50–100) |
| Gabapentin | 38 (0–63) | 50 (0–93) | 60.5 (21–90) | 50 (27–94) |
| Opiates | 75 (64–85) | 68 (31–99) | 26 (0–70) | 28 (0–96) |

The table gives the median and (range) for how often panel members use the stated drugs to treat PSH, on a continuous scale where 0=never, 50=half of patients, 100=always. Note: Data characteristics suggested technical difficulties for two panel participants: in one case '50' was registered for all drugs in all care settings, which seems unlikely to be the intended response. In another case 0 was registered for all drugs for all care settings after intermediary care; the instruction was to not answer for care settings that respondents do not work in rather than to answer 0. These responses have been excluded from the summary above.
PSH, paroxysmal sympathetic hyperactivity.

## Choice of drugs for PSH treatment

In round 1, the panel indicated which drugs they use with a primary purpose of treating PSH. To support recall of any less usual choices, and to aid analysis, panel participants were presented with a list of 29 drug classes with examples from a 2021 Scoping Review on PSH[4] and asked to indicate all drugs they used: there was an unlimited number of choices, and no restrictions were made at this stage regarding phase of recovery. Drugs used were (drug, number of panel participants): Opiates (15) non-selective β-blockers, for example, Propranolol (14) Gabapentionoid, for example, Gabapentin (14) GABA-B agonist, for example, Baclofen (11), α2 agonist, for example, Clonidine (10), Paracetamol (10), selective β-blockers (4), antiepileptics, for example, Carbamazepine, Valproate (3), antihistamine, for example, Hydroxyzine (3), Benzodiazepines, for example, Diazepam (3), Other anaesthetics, for example, Propofol (3), Adamantane, for example, Amantadine (2), Sedative hypnotics, for example, Zopiclone (2), Barbiturates, for example, Phenobarbital (1), Dopamine agonist, for example, Bromocriptine (1) Hyrdrazinophalazine, for example, Didhydralazine (1), Ibuprofen (1), Other—Muscle relaxant (1), Others (0).

In round 2, panel participants indicated how often they used the drugs that had been most commonly cited in round 1, for each of the phases of care in which they were involved (see table 1), by moving a horizontal slider for each drug. The five drug classes for which responses were sought, and given in the table, were those which in round 1 were included in at least 25% of individual panel participants' lists of their three most used drugs, at any stage of care.

## Algorithm

The consensus statements and panel participants' drug choices are summarised in the algorithm in figure 1, which is intended to support clinicians in their assessment and management of patients with PSH. The algorithm was presented to the panel in round 3 with resulting comments incorporated into the final version.

## DISCUSSION

After an initial exploratory round in this Delphi process on the management of PSH during rehabilitation after severe acquired brain injury, consensus was reached for 18 statements in round 2, on terminology, assessment, non-pharmacological management and treatment principles. Consensus for three further statements and refined wording for two statements were achieved in round 3.

Responses on drug choices were less clear cut: panel participants' drug choices at the various phases of care show clear trends, but also a considerable variability in practice. This was not unexpected given the weak state of the evidence.[4] Alpha-2 agonists and opiates are used mostly in the early stages of recovery, on the neurointensive care and intermediary care units, and infrequently later during inpatient care and rehabilitation, although some panel participants continue to use these drugs even at these later stages. Non-selective beta-blockers are most commonly used at all phases of care after neurointensive care, and Gabapentin is also commonly used in these phases.

These drug choices are consistent with targets suggested by current theoretical models[5 7] that focus on failure of central inhibition of sympathetic and motor responses,

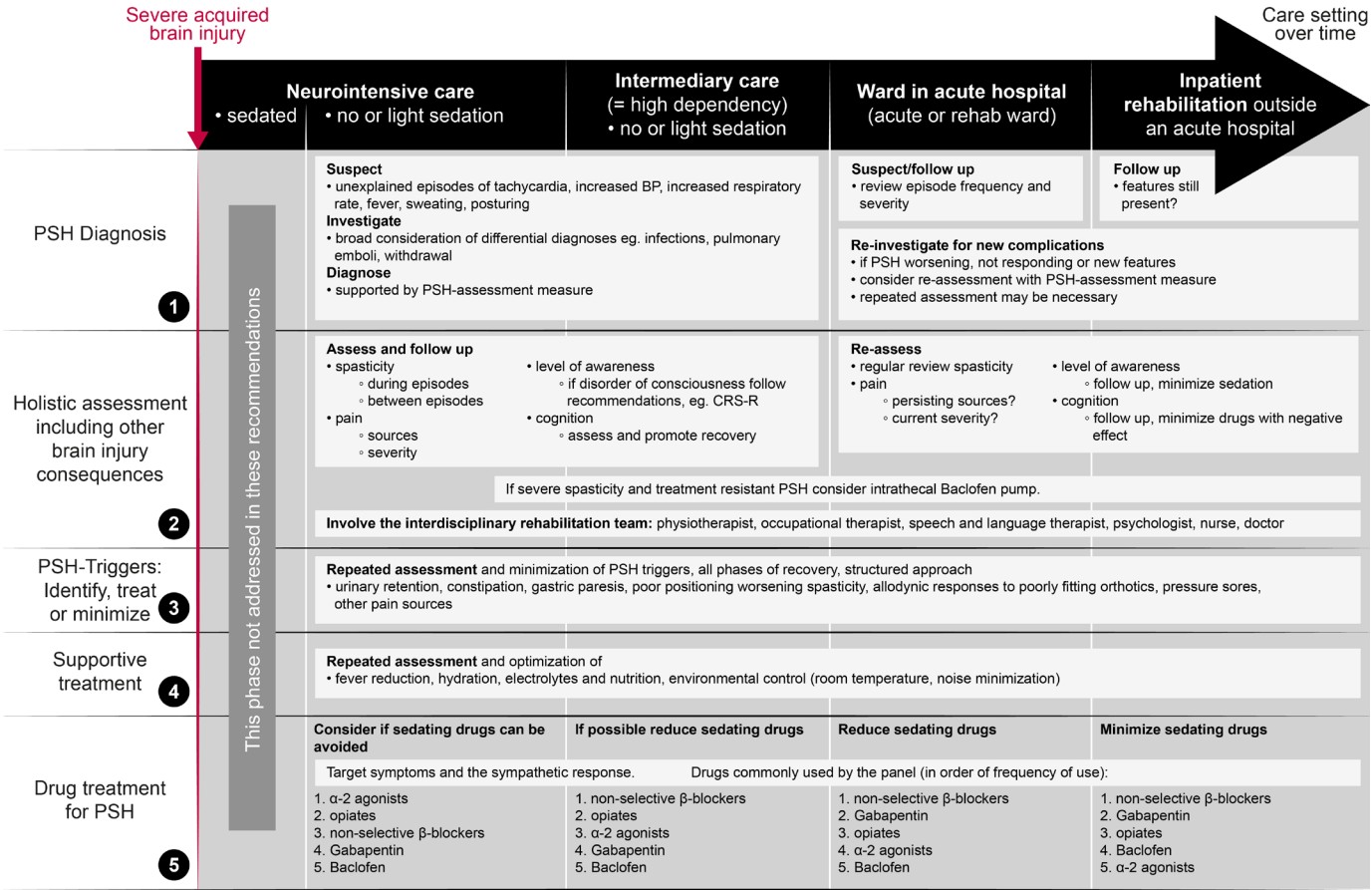

**Figure 1** PSH after severe acquired brain injury–algorithm for assessment and management as part neurorehabilitation. CRS-R, coma recovery scale-revised; PSH, paroxysmal sympathetic hyperactivity.

mediated by high levels of catecholamines during episodes. Alpha-2 agonists such as Clonidine reduce sympathetic outflow but also cause sedation. Opiates are effective against pain responses, but unwanted effects including sedation, respiratory depression and cognitive effects become relevant as rehabilitation progresses. The non-selective beta-blocker propranolol is both cardio-protective and neuroprotective, reduces the effects of high catecholamine levels and reduces metabolic rate.[15] A recent single-centre randomised controlled study of propranolol and clonidine for severe traumatic brain injury however found no effect on outcome;[16] this study was however not restricted to patients with PSH so the relevance of the finding is unclear. Gabapentin acts on voltage-dependent calcium channels in the dorsal horns of the spinal cord to inhibit neuropathic pain,[17] and as such may decrease the uncontrolled nociceptive or allo-dynic drive to PSH, and can be safely used without contin-uous monitoring.

Trials to date are limited and do not give sufficient evidence on which to base specific drug choices during an individual patient's recovery and rehabilitation; the consensus statements developed through this Delphi process likely represent clinical syntheses by the panel of a broad range of parameters related to the holistic reha-bilitation process in coming to decisions about positive

and negative aspects of particular drugs, in the face of insufficient evidence.

Several drugs on the list provided to panel members from the scoping review[4] were used by no or only one panel member; generally these were, as expected, drugs with the weakest of evidence of and/or putative mecha-nisms for effect, for example, Hydralazine, Angiotension 2 antagonists and Phenothiazines. However, the dopa-mine agonist Bromocriptine, considered to target hyper-pyrexia and sweating, with a reasonable theoretical basis and more extensive (although still weak) evidence base was also only used by a single panel member. Reasons for this non-use of Bromocriptine were not further investi-gated, but could include side effect profile, safety consid-erations in care settings without continuous monitoring, and/or lack of familiarity.

Panel feedback also highlighted certain management difficulties due to non-patient factors. For example, although consensus was reached that an intrathecal baclofen pump should be considered for the most severe, refractory cases, this may be unavailable in some centres. While intrathecal baclofen is well established for the treat-ment of severe spasticity without PSH, and case reports and series exist dating back 20 years (see reference[18] for summary), the quality of evidence remains poor for this potentially effective but invasive intervention.

A number of the consensus statements are of a confirmatory nature, reiterating key aspects of the literature on PSH and putting these into a clinical context for routine practice. For example, although it will soon be 10 years since the publication of consensus on terminology and diagnosis, initial discussions and feedback to round 1 showed that other terms remain in use in today's clinical practice in the Scandinavian countries. Some statements also make explicit aspects that might be considered obvious components of good care, but where panel particpants' clinical experiences are of common difficulties (eg, non-tapering of drugs such as PSH abates). It is hoped that this broad Scandinavian consensus on recommended terminology, assessment and treatment principles will support good clinical practice for patients with PSH. Other consensus statements begin to address common difficulties encountered in later clinical management of patients who may need to be transferred between units several times before leaving inpatient care. The panel's experience is that indications for each prescribed drug are not always specified at the time of transfer between units, which creates difficulties in follow-up and ongoing optimisation of drug treatment.

The consensus process was guided by the initial exploratory round and focused on areas prioritised during neurorehabilitation. Other areas of clinical uncertainty exist where further guidance would aid clinicians. One example is difficulties in applying current diagnostic criteria when suspicion of PSH coexists with a (suspected or diagnosed) complication that could in part explain PSH-like symptoms. PSH is currently a diagnosis of exclusion,[1] and the PSH-AM does not allow for such a clinical scenario.

Strengths of the consensus process include the following of published recommendations for Delphi processes, an explorative first round, clear differentiation in questions to panel participants of the various phases of care (from neurointensive care to a rehabilitation ward outside an acute hospital), inclusion of panel participants from several medical specialities, inclusion of non-pharmacological aspects of assessment and management, and broad geographical representation from Denmark, Norway and Sweden. Limitations include an insufficient number of panel participants with experience of care after discharge from inpatient rehabilitation for consideration of this phase, and lack of data on the extent of panel participant's clinical experience of PSH; although it was a requirement that panel participants had been involved in clinical decisions for patients with PSH during the past year, there was no specific requirement regarding the number of patients treated by panel participants. We acknowledge difficulties in defining the start point for these recommendations; the description used, 'breathing spontaneously with no or light sedation on the neurointensive care unit', was developed to portray concisely to the panel a patient who has survived initial neurosurgical interventions and is entering a phase where rehabilitation considerations such as recovery of awareness and cognition begin to become relevant and where sedating drugs may potentially adversely affect such processes. We did not however further define 'light sedation'.

The limitations of a Delphi process representing expert opinion should also be noted: studies with randomised, controlled methodologies are needed to provide evidence-based recommendations. These should be based as far as possible on knowledge of underlying PSH-mechanisms, and ideally include patients during all stages of care, with long-term follow-up. However, given the clinical complexity of patients with PSH, logistical and ethical barriers to such studies are considerable.

We consider that this Delphi process adds to previous studies by giving more explicit consideration to changing priorities through the rehabilitation chain of care and clarifies the necessity of an interdisciplinary approach to assessment and management involving the rehabilitation team, at all phases including very early rehabilitation. Identifiable triggers were noted in a high proportion (72%) of patients in one study.[7] The panel's recommendation of a systematic interdisciplinary approach to identifying and minimising these triggers seems justified.

An algorithm has been included (see figure 1) as a summary of the consensus statements, with the intention of assisting physicians in treatment of patients with PSH in all phases of inpatient care after early sedation on the neurointensive care unit has been reduced.

While robust research studies of the effects of drugs on PSH are desirable, the clinical complexity of most patients affected by PSH and its relative rarity are expected to make definitive studies difficult to perform. Studies evaluating real-world data from treatment of patients carried out during routine clinical practice are also needed to improve treatment. For this to be possible, better structures are needed for assessment, diagnosis, treatment and follow-up. 'Structures' include routines guided by these consensus criteria, and also consideration of medical journal systems and coding; there is no ICD-10 diagnosis for PSH which is a hinder to data assimilation from clinical practice.

One problem with current treatment strategies is that they are of necessity reactive. Ideally early, subclinical signs of PSH would be identified. Treatment of such preclinical stages would have theoretical benefits. Mathematical modelling of continuous patient data[6] is an area of emerging interest together with machine learning and has the potential to aid early identification and treatment of PSH. Extending such methods to phases of care where continuous vital parameter monitoring is no longer available is a further challenge.

There are also examples of limitations in electronic medical journal systems preventing documentation of PSH as a drug indication as the term has more characters than allowed in some prescribing systems. Such organisational and system factors should receive more attention in future studies and planning for care of patients with severe acquired brain injury. Continued educational efforts about PSH are also needed to inform the less

specialised healthcare staff that the patient is likely to encounter with increased time after brain injury.

## CONCLUSION

A considerable degree of consensus has been reached in Scandinavian countries for the clinical assessment and management of patients with PSH after severe acquired brain injury. Uncertainties remain about specific drug choices, which need to be modified in pace with changing rehabilitation priorities.

**Acknowledgements** We thank the following: Doctors Per Ertzgaard, Rehabilitation Physician and Fredrik Ginstman, Neurosurgeon, both of the University Hospital Linköping, Sweden, for valuable comments on the project plan and a pilot version of the questions for round 1. The panel participants for their valuable contributions to the Delphi Consensus process: Daniel Dahlgren and Alison Godbolt, Department of Rehabilitation Medicine, Danderyd Hospital, Sweden. Sofie Jacobson, Department of Rehabilitation Medicine, Umeå University Hospital, Sweden. My Dung Nguyen Torkildsen, Head of department, Physical medicine and rehabilitation, Stavanger University Hospital, Norway. Tanja Bertheussen, Department of Physical Medicine and Rehabilitation, Kongsgård, Sørlandet Hospital, Norway. Alba Corell, Department of Clinical Neuroscience, Institute of Neuroscience and Physiology, Sahlgrenska Academy, University of Gothenburg, Gothenburg, Sweden, Department of Neurosurgery, Sahlgrenska University Hospital, Gothenburg, Sweden. Kristin Alvsaaker, Postoperative and Intensive Care Department, Oslo University Hospital (OUH), Norway and Department of Physical Medicine and Rehabilitation, OUH, Norway. Tiina Ader, Department of Physical Medicine and Rehabilitation, Haukeland University Hospital, Norway. Merete Petersen, Department of Neuroanesthesiology, Rigshospitalet, University Hospital, Denmark. Cathrine Elisabeth Einarsen, Clinic of Rehabilitation, St. Olavs Hospital, Trondheim University Hospital, Trondheim, Norway. Tove Grønbæk Jensen, Hammel neurocentre, Denmark. Marina Parziali, Neurorehabilitation, NU Health Care, Sweden. Ursula Heldmann, Rehabilitation medicine, Skane University Hospital, Sweden. Astrid Wille-Jørgensen, Department of Brain and Spinal Cord Injury, Bodil Eskesen Center, Rigshospitalet, Demark. Maritza Beekman, Department of Rehabilitation Medicine, University Hospital Linköping, Sweden. Marco Brizzi, Senior Consultant Neurologist, Neuro Intermediate Care Unit, Malmoe, Dept of Neurology and Rehabilitation medicine, Skane University Hospital, Sweden. Caroline Ustvedt, Department of Traumatic Brain Injury, Sunnaas Rehabilitation Hospital, Norway. Shirin Kordasti Frisvold, Chief consultant, Department of Intensive Care and Department of Anesthesiology and Intensive Care, University Hospital of North Norway, Tromso. Rasmus Langelund Jørgensen, Hammel Neurorehabilitation and Research Centre, Denmark. Laura Serrano Barrenechea, Department of Rehabilitation Medicine South Älvsborgs Hospitla and Ryhov Hospital. Stefan Arousell, Dept of Rehabilitation Medicine, Danderyd hospital, Stockholm for graphic design support in creation of the figure.

**Contributors** All authors contributed to study design, conduction of the study and analysis of data. Drs AZ and CND formed and distributed the surveys in REDCap. Dr AKG drafted the article and created a sketch for the figure. Drs AZ and CND critically appraised and approved the article text. Dr AKG is the guarantor.

**Funding** The authors have not declared a specific grant for this research from any funding agency in the public, commercial or not-for-profit sectors.

**Competing interests** Dr AKG is the specialty expert for Rehabilitation Medicine for the Stockholm Region. Dr AZ has received compensation from Ipsen for support with a workshop at the Nordic Toxins conference, October 2023. Dr CND is the Chair for the National Program Group, and for the Regional Program Group for Stockholm-Gotland, for Rehabilitation Habilitation and Insurance Medicine, within the National system for knowledge-driven management within Swedish healthcare.

**Patient and public involvement** Patients and/or the public were not involved in the design, or conduct, or reporting or dissemination plans of this research.

**Patient consent for publication** Not applicable.

**Ethics approval** This study involves human participants. The participants were physician colleagues who participated in a Delphi consensus process, that is, they were not patients. Ethical approval was not required. Participants gave informed consent to participate in the study before taking part.

**Provenance and peer review** Not commissioned; externally peer reviewed.

**Data availability statement** Data are available upon reasonable request. Deidentified data are available upon reasonable request to the corresponding author on presentation of a valid research question from an established researcher.

**ORCID iDs**
Alison K Godbolt http://orcid.org/0000-0003-4511-6181
Alexandros Zampakas http://orcid.org/0009-0004-9765-3725
Catharina Nygren Deboussard http://orcid.org/0009-0000-0289-0527

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
