## [Reviewer comments · BMJ Open]

ARTICLE DETAILS

TITLE (PROVISIONAL)	Paroxysmal Sympathetic Hyperactivity during Neurorehabilitation for severe acquired brain injury: Current Scandinavian Practice and Delphi Consensus Recommendations
AUTHORS	Godbolt, Alison; Zampakas, Alexandros; Nygren Deboussard, Catharina

VERSION 1 – REVIEW

REVIEWER	Podell, Jamie University of Maryland School of Medicine
REVIEW RETURNED	11-Feb-2024

GENERAL COMMENTS	In this article, authors report the results of a Delphi Consensus process (including 20 clinicians from Nordic countries involved in acute brain injury patient care during the rehabilitative phase) for making recommendations regarding diagnosis and treatment strategies for paroxysmal sympathetic hyperactivity (PSH). This is a nice article on an important topic that will provide useful guidance, especially for clinicians who do not frequently encounter PSH. The methods appear valid and easy to understand, and the resulting recommendations nicely incorporate both the usefulness of the PSH-AM consensus paper and some of its remaining unanswered questions in a practical manner. Minor issues: I think the scope of the recommendations and treatment phases could be better defined with reference to the goals of treatment. Specifically, I think that the description of the earliest phase of care (to which these recommendations are said not to apply) would be better defined by treatment goals than by level of sedation for a number of reasons. For one, duration and depth of sedation varies greatly by institutional and individual practice patterns and by specific type of acute brain injury. This could be a major source of confusion for readers, especially for providers who may practice in an ambiguous phase of care based on this guideline. For example, an ICU provider who does not use deep sedation even in the hyperacute phase of ABI, when more aggressive management of PSH may be warranted for rational mechanistic reasons. In my opinion, this earliest phase of care might be better defined by that time when patients are deemed at highest risk for secondary brain injury, when more aggressive approaches to hemodynamic fluctuations, fever, and other PSH-relevant sources of morbidity are justified. Regarding the recommendations themselves, I take issue with some of the medication preferences but acknowledge lack of evidence and practice variation. In particular, my institution prefers
---

	bromocriptine, a dopamine agonist, (in addition to propranolol) as a first line agent in most instances of PSH. This is because of its effect (albeit with limited, though increasing, evidence) on central fevers. I am surprised that this medication did not make it onto the authors' list and would be interested in authors' commentary on this. My own thoughts are that, in the ICU where I practice, we may be more concerned about fever as a source of secondary brain injury in the acute phase. Alternatively, central fevers may become less apparent in the subacute/rehabilitative phase to which these recommendations apply? This relates to my above point regarding a more mechanistic and rational explanation of the treatment phases and scope of recommendations. Additionally, while we occasionally use gabapentin, I have found this medication to be less effective for PSH in the ICU. Were specific doses of these medications considered as part of the recommendations? Lastly, there are a few diagnosis and management questions that were not addressed by these recommendations that come to mind for me. For example, what are common alternative diagnoses for clinicians to consider? Is there a consensus on how to manage PSH-like symptoms in the presence of a possible alternative underlying diagnosis (e.g. infection)? What should trigger consideration of important alternative diagnoses after a diagnosis of PSH has been established to avoid anchoring bias? Even with the use of the PSH-AM, uncertainty regarding a diagnosis of PSH often persists. Did the panel suggest specific PSH-AM cutoffs or allowing for a certain duration of time to pass to "establish a diagnosis"?
--	--

REVIEWER	Ljungqvist, Johan Sahlgrenska Sjukhuset, Neurosurgery
REVIEW RETURNED	16-Feb-2024

GENERAL COMMENTS	The authors present an interesting paper on the management of PSH in TBI-patients across Scandinavia. However, a major problem is that PSH is often missed or under-diagnosed, but the authors hereby raise awareness and provide a practical guide to other clinicians about management. The Delphi process seems adequate to address the issue of writing guidelines in lack of evidence, but a deeper scientific background to each statement would enhance the paper. The list of references is short and refer mainly to other reviews. Perhaps, this paper could be published along with a separate review of the literature by the authors? The authors have clearly stated the main limitations of the paper; in the abstract as well as in the text. The main result, that PSH is treated similarly (among the interested specialists in the study) in Scandinavia, makes a limited contribution to the scientific literature, but the practical consensus statements (and the flowchart) about treatment will be appreciated. The authors refer to a "Nordic Consensus" but only the Scandinavian countries are considered, hence "Scandinavian" would be more appropriate
--

VERSION 1 – AUTHOR RESPONSE

Responses to Reviewers' comments:

Responses to reviewer 1. Line numbers refer to the marked copy.

Minor issues:

I think the scope of the recommendations and treatment phases could be better defined with reference to the goals of treatment. Specifically, I think that the description of the earliest phase of care (to which these recommendations are said not to apply) would be better defined by treatment goals than by level of sedation for a number of reasons. For one, duration and depth of sedation varies greatly by institutional and individual practice patterns and by specific type of acute brain injury. This could be a major source of confusion for readers, especially for providers who may practice in an ambiguous phase of care based on this guideline. For example, an ICU provider who does not use deep sedation even in the hyperacute phase of ABI, when more aggressive management of PSH may be warranted for rational mechanistic reasons. In my opinion, this earliest phase of care might be better defined by that time when patients are deemed at highest risk for secondary brain injury, when more aggressive approaches to hemodynamic fluctuations, fever, and other PSH-relevant sources of morbidity are justified.

Response: We agree that defining phases of care after severe brain injury and relating these to PSH-treatment goals and rehabilitation priorities is an area of difficulty. There are no clear cut, established definitions for phases of care; at the same time ambiguity regarding care setting and phase of recovery risks confusion about treatment possibilities (including safety in medical settings without continuous monitoring) and priorities. This article focuses on PSH treatment during neurorehabilitation; we therefore developed a pragmatic definition of the start point for neurorehabilitation in discussion with a neurosurgeon. With a rehabilitation focus and need for consideration of factors such as impact of PSH-treatment on conscious level and cognition, an appropriate start point for this consensus process was from the time point at which the patient is awake, when such considerations become relevant. Complete exclusion of all forms of sedating medication would have risked excluding some patients in an early rehabilitation phase, hence the formulation of the pragmatic description “light sedation” as a compromise. We thank the reviewer for the suggestion of an alternative definition of the earliest phase of care, not covered by this article, which could be considered in future work.

Changes and additions have been made to the paragraph on “rehabilitation focus” in the methods, to make this clearer (lines 140 - 147) , and an addition has also been made to the limitations section of the discussion (lines 421 - 427) to address this.

Regarding the recommendations themselves, I take issue with some of the medication preferences but acknowledge lack of evidence and practice variation. In particular, my institution prefers bromocriptine, a dopamine agonist, (in addition to propranolol) as a first line agent in most instances of PSH. This is because of its effect (albeit with limited, though increasing, evidence) on central fevers. I am surprised that this medication did not make it onto the authors' list and would be interested in authors' commentary on this. My own thoughts are that, in the ICU where I practice, we may be more concerned about fever as a source of secondary brain injury in the acute phase. Alternatively, central fevers may become less apparent in the subacute/rehabilitative phase to which these recommendations apply?

Response: Thank you for the comment. We agree that practice variations are to be expected when evidence is lacking. There was in fact one panel member who used dopamine agonists – information on this is given in the section on “Choice of Drugs for PSH-treatment” in the results. The example given to panel members within the drug class “dopamine agonists” was Bromocriptine. In the initial submission we listed examples for some but not all classes of medications in this section, to keep this section concise, and the example Bromocriptine was not specifically mentioned. **To aid**

clarity and improve consistency we now include all the drug examples given to the panel after each class of medication (lines 328 – 333).

We also agree that of all the drugs used by no or only one panel member, Bromocriptine could reasonably be highlighted due to the reasonable theoretical mechanisms and relatively frequent mention in the literature. **We have therefore added a paragraph to the discussion on this (lines 378-385).**

This relates to my above point regarding a more mechanistic and rational explanation of the treatment phases and scope of recommendations.

Response: We have added a paragraph to the introduction to summarize the state of knowledge about underlying mechanisms and treatment targets (lines 67-81). We do agree with the reviewer that a greater understanding of underlying mechanisms would be a major step forward in developing evidence-based care and support current and future research efforts. The aim of this consensus process is to improve care for today's patients, until such evidence exists. This rationale is discussed in the introduction.

Additionally, while we occasionally use gabapentin, I have found this medication to be less effective for PSH in the ICU. Were specific doses of these medications considered as part of the recommendations?

Response: Specific doses of Gabapentin were not considered as such a degree of detail lay outside what the consensus process could reasonably manage. We agree that such details are of interest for future work.

Lastly, there are a few diagnosis and management questions that were not addressed by these recommendations that come to mind for me.

For example, what are common alternative diagnoses for clinicians to consider?

Response: As you know, Baguley (ref 1) gives examples. **We have added text on this to the introduction (lines 56-61).**

Is there a consensus on how to manage PSH-like symptoms in the presence of a possible alternative underlying diagnosis (e.g. infection)?

Response: This was not considered in this consensus process, but we agree that this can be a clinical problem. **We have added a section to the discussion on this (lines 405 – 410).**

What should trigger consideration of important alternative diagnoses after a diagnosis of PSH has been established to avoid anchoring bias?

Response: Information on this is given in the algorithm (figure); "PSH worsening, not responding, or new features).

Even with the use of the PSH-AM, uncertainty regarding a diagnosis of PSH often persists. Did the panel suggest specific PSH-AM cutoffs or allowing for a certain duration of time to pass to "establish a diagnosis"?

Response: As you know the PSH-AM in published form includes cut-offs for unlikely/possible/probable PSH. We did not focus on further details related to the PSH-AM as at the commencement of the consensus process only 9 panel members used the PSH-AM.

Responses to reviewer 2: Line numbers refer to the marked copy.

The authors present an interesting paper on the management of PSH in TBI-patients across Scandinavia. However, a major problem is that PSH is often missed or under-diagnosed, but the authors hereby raise awareness and provide a practical guide to other clinicians about management.

The Delphi process seems adequate to address the issue of writing guidelines in lack of evidence, but a deeper scientific background to each statement would enhance the paper. The list of references is short and refer mainly to other reviews. Perhaps, this paper could be published along with a separate review of the literature by the authors?

Response: Thank you for the suggestion of another review. Understanding of PSH is unfortunately still largely at the level of insufficiently proven hypotheses which are summarized in already published reviews (for example the 2017 Lancet Neurology review), and reports of treatment options without sufficient evidence for effect. We note that a new review published parallel to this study (after the first two rounds of this consensus process, *Neurol Ther.* 2024 Feb;13(1):11-20. doi: 10.1007/s40120-023-00561-x. Paroxysmal Sympathetic Hyperactivity After Acquired Brain Injury: An Integrative Review of Diagnostic and Management Challenges, Xu, Zhang, Li) did not identify any important new evidence, so we do not consider it would give any added value for us to do another review at present.

We do agree that more information on the state of current knowledge would enhance the paper and have therefore made additions to the introduction (lines 67-81) and discussion (lines 360 – 372) regarding pathophysiological aspects.

The authors have clearly stated the main limitations of the paper; in the abstract as well as in the text. The main result, that PSH is treated similarly (among the interested specialists in the study) in Scandinavia, makes a limited contribution to the scientific literature, but the practical consensus statements (and the flowchart) about treatment will be appreciated.

The authors refer to a "Nordic Consensus" but only the Scandinavian countries are considered, hence "Scandinavian" would be more appropriate

Response: We agree that “Scandinavian” is more geographically correct as a description of the working locations of panel members. **References to “Nordic” have been changed to “Scandinavian” where appropriate.**

VERSION 2 – REVIEW

REVIEWER	Podell, Jamie University of Maryland School of Medicine
REVIEW RETURNED	24-Apr-2024
GENERAL COMMENTS	Again, this is a very nice summary of consensus-based recommendations for assessment and treatment of PSH in the rehab setting. I am satisfied by the authors' commentary